# State of the Art of Antimicrobial and Diagnostic Stewardship in Pediatric Setting

**DOI:** 10.3390/antibiotics14020132

**Published:** 2025-01-27

**Authors:** Daniele Donà, Elisa Barbieri, Giulia Brigadoi, Cecilia Liberati, Samantha Bosis, Elio Castagnola, Claudia Colomba, Luisa Galli, Laura Lancella, Andrea Lo Vecchio, Marianna Meschiari, Carlotta Montagnani, Maia De Luca, Stefania Mercadante, Susanna Esposito

**Affiliations:** 1Division of Pediatric Infectious Diseases, Department for Woman and Child Health, University of Padua, 35128 Padua, Italy; daniele.dona@unipd.it (D.D.); elisa.barbieri@unipd.it (E.B.); giulia.brigadoi@gmail.com (G.B.); cecilia.liberati@unipd.it (C.L.); 2Pneumology and Infectious Diseases Unit, Fondazione IRCCS Ca’ Granda Ospedale Maggiore Policlinico, 20122 Milano, Italy; samantha.bosis@gmail.com; 3Infectious Diseases Unit, Department of Pediatrics, IRCCS Istituto Giannina Gaslini, 16147 Genoa, Italy; eliocastagnola@gaslini.org; 4Division of Pediatric Infectious Diseases, “G. Di Cristina” Hospital, ARNAS Civico Di Cristina Benfratelli, University of Palermo, 90134 Palermo, Italy; claudia.colomba@unipa.it; 5Department of Health Sciences, University of Florence, 50139 Florence, Italy; luisa.galli@unifi.it; 6Infectious Diseases Unit, Meyer Children’s University Hospital, IRCCS, 50139 Florence, Italy; carlotta.montagnani@meyer.it; 7Infectious Diseases Unit, Bambino Gesù Children’s Hospital, IRCCS, 00165 Rome, Italy; laura.lancella@opbg.net (L.L.); maia.deluca@opbg.net (M.D.L.); stefania.mercadante@opbg.net (S.M.); 8Department of Translational Medical Sciences, Federico II University, 80138 Naples, Italy; andrea.lovecchio@unina.it; 9Department of Infectious Diseases, Azienda Ospedaliero-Universitaria di Modena, University of Modena and Reggio Emilia, 41125 Modena, Italy; mariannameschiari1209@gmail.com; 10Pediatric Clinic, Parma University Hospital, Department of Medicine and Surgery, University of Parma, 43126 Parma, Italy

**Keywords:** antibiotics, antimicrobial resistance, antimicrobial stewardship programs, pediatric infectious diseases, point-of-care tests, rapid diagnostic tests

## Abstract

Antimicrobial stewardship programs (ASPs) and diagnostic stewardship programs (DSPs) are essential strategies for effectively managing infectious diseases and tackling antimicrobial resistance (AMR). These programs can have a complementary impact, i.e., ASPs optimize antimicrobial use to prevent resistance, while DSPs enhance diagnostic accuracy to guide appropriate treatments. This review explores the current landscape of ASPs and DSPs in pediatric care, focusing on key factors, influencing their development, implementation, and evaluation across various settings. A multidisciplinary approach is necessary, involving multiple healthcare professionals to support comprehensive stewardship practices in pediatric care. No single intervention suits all settings, or even the same setting, in different countries; interventions must be tailored to each specific context, considering factors such as hospital capacity, patient complexity, and the parent–child dynamic. It is essential to educate caregivers on optimal antibiotic use through clear, concise messages adapted to their socioeconomic status and level of understanding. The cost-effectiveness of ASPs and DSPs should also be assessed, and standardized metrics should be employed to evaluate success in pediatric settings, focusing on outcomes beyond just antibiotic consumption, such as AMR rates. This manuscript further discusses emerging opportunities and challenges in ASP implementation, offering insights into future research priorities. These include large-scale studies to evaluate the long-term impact of ASPs, cost-effectiveness assessments of pediatric-specific diagnostic tools, and the integration of artificial intelligence to support clinical decision making. Addressing these areas will enhance the effectiveness and sustainability of ASPs, contributing to global efforts to combat AMR and improve pediatric health outcomes.

## 1. Introduction

Programs focused on the appropriate use of antimicrobials and diagnostic tools are essential for managing infectious diseases effectively and efficiently [1]. Antimicrobial stewardship programs (ASPs) have become a cornerstone in the fight against antimicrobial resistance (AMR), a critical public health issue also recognized globally in the pediatric population [1,2,3].

One of the primary goals of pediatric ASPs is to reduce the inappropriate use of antibiotics, which remains a significant challenge [1]. Children are frequently prescribed antibiotics, especially in outpatient settings, and unnecessary use or suboptimal doses are common [4,5,6], leading to potential adverse drug reactions and the emergence of resistant organisms, which can complicate future treatment options and lead to poorer health outcomes [7]. Moreover, the developing immune system and microbiota in children can be profoundly affected by antimicrobial exposure, with long-term health implications including allergies, asthma, and obesity [8,9,10,11].

In pediatrics, these programs should rely on the best available evidence, ideally from studies specifically conducted in children, considering the unique aspects related to age, growth, and physiology. However, despite significant strides made in adult healthcare, there is a relative paucity of literature and consensus on the best practices tailored for pediatric settings [12]. Indeed, the implementation of ASPs in pediatric settings requires taking into account numerous additional factors, such as, for example, parent anxiety and pressure to prescribe antibiotics, especially in outpatient care. Prescribers’ decisions may be influenced by emotional factors, for example, fear, diagnostic uncertainty, anxiety, perceived risks, and challenges in communication between doctors and parents. Furthermore, the pediatric population presents unique challenges and opportunities for ASPs due to differences in pharmacokinetics, disease epidemiology, and the impact of antimicrobial use on developing microbiomes [4,13,14].

Successful implementation of pediatric ASPs requires a multidisciplinary approach, not only via collaboration between different healthcare workers but also by engaging with patients and their families through education in order to reduce unnecessary antibiotics use and increase compliance [15]. This narrative review provides an overview of the current knowledge on ASPs and diagnostic stewardship programs (DSPs) in pediatric settings based on the existing literature, expert opinions, and case studies. We will focus on the definition of ASPs and DSPs, the different healthcare workers who should be involved, the different types of ASPs and DSPs, and their implementation in different pediatric settings, considering inpatient settings, outpatient settings, and pediatric emergency departments (PEDs). Moreover, we will focus on the cost-efficacy of these programs and on the different metrics that should be used to evaluate the efficacy of the implementation of an ASP or DSP in pediatric care, considering not only antibiotic consumption but also other outcomes, such as, for example, antibiotic resistance rate. Finally, we will focus on the different steps for the creation and implementation of these programs and on the challenges related to their implementation and sustainability.

## 2. What Is an Antibiotic Stewardship Program (ASP) and a Diagnostic Stewardship Program (DSP)?

An ASP is a comprehensive set of actions designed to promote the responsible use of antibiotics [16,17]. This involves selecting the appropriate drug, dose, timing, frequency, and route of administration based on knowledge of the antimicrobial spectrum and pharmacokinetic/pharmacodynamic data, ideally tailored to the specific clinical condition, limiting the risk of resistance development and the risk of side effects, without compromising the patient outcome.

A DSP encompasses activities that complement an ASP by ensuring the appropriate use of diagnostic tests in clinical settings to guide appropriate treatment [18,19]. It aims to direct testing towards the appropriate patients, thereby improving the correct use of antibiotics. This includes selecting the most suitable microbiological tests for the clinical situation, executing these tests correctly, and choosing the right tests to assess the patient’s initial clinical conditions and response to treatment. The proper execution and evaluation of antibiotic sensitivity tests are fundamental for managing infectious diseases. Additionally, evaluating the plasma (and potentially tissue) concentrations of drugs, when available, can further support the optimal management strategy.

ASPs and DSPs should be implemented in a coordinated and parallel approach, as they are closely interrelated and have a synergistic effect. Together, they represent two steps of the same process, with the ultimate goal of optimizing antibiotic prescriptions, e.g., reducing the unnecessary use of broad-spectrum antibiotics while ensuring their appropriate use in empirical treatment, when needed.

## 3. Which Healthcare Workers Should Be Involved in an ASP? The Antimicrobial Stewardship Team

Every ASP requires a multidisciplinary team tasked with defining governmental policies for the responsible use of antibiotics, aligning with infection control protocols [20,21]. The Antimicrobial Stewardship Team (AST) in pediatric care should be led by a pediatric infectious disease specialist or, when absent, by a pediatrician with expertise in antibiotic therapy. This role is essential for ensuring that treatment recommendations and guidelines are grounded in the latest evidence in the field. Including a physician specializing in pediatric infectious diseases strengthens the program’s credibility. Direct engagement with prescribing physicians and the clinical governance committee is crucial; interaction with the former helps educate them on improved prescribing practices and fosters collaboration in discussing challenging cases or barriers faced during the program’s implementation; engagement with the latter is necessary to secure formal approval and financial support for the program’s execution [22].

Additionally, the team should also include at least one expert from each relevant field, i.e., a nurse, a clinical pharmacist, a clinical microbiologist, a public health specialist, dietitians, and other physicians, depending on the wards involved in the implementation. Each team member should have clearly defined roles [16,23]. Although nurses are not formal prescribers in most countries, they play a crucial role in antimicrobial communication and management, actively engaging in encouraging, reminding, and overseeing prescribers’ choices, while questioning decisions, when necessary. Moreover, they are responsible for preparation, evaluation of incompatibility with other drugs administrated at the same time, and infusion time. Their involvement has shown significant potential in optimizing antimicrobial use, especially in areas such as monitoring the choice of antimicrobial, timing, therapy duration, and dosing. Clinical pharmacists play an important role in promoting the rational use of antimicrobial drugs and educating physicians; their competency regarding new available drugs and possible adverse events or possible interactions with other drugs plays a crucial role, especially in complicated patients in more fragile settings. Clinical microbiologists are essential, not only in the management of acute cases, during which they could provide crucial information regarding the type of pathogen and the spectrum of resistance to target antibiotic therapy, but also in the choice of which antimicrobials should be limited or reassessed in a specific setting. Indeed, they should provide, on a regular basis, the epidemiology of the hospital, with the recent burden of antibiotic resistance in the different wards, to allow for updating recommendations on empirical antibiotic therapy [22]. Especially in patients receiving parenteral nutrition (TPN), dietitians can help evaluate interaction and incompatibility with TPN when administered simultaneously and through the same route. The involvement of physicians from the ward where the program is implemented is essential to establish a bridge between the AST and the ward team, fostering cooperation and collaboration across both groups. The multidisciplinary nature of the AST ensures a comprehensive and integrated approach to optimal antibiotic treatment management (Figure 1).

## 4. Different Types of ASPs

ASP interventions can be classified as core strategies (i.e., pre-authorization of restricted antimicrobials and prospective audit and feedback), minor interventions [16], or interventions before and after the prescriptions [24]. According to the Infectious Diseases Society of America and the Society for Healthcare Epidemiology of America (IDSA/SHEA) guidelines, programs should determine whether to include one core strategy or a combination of both strategies, based on the availability of facility-specific resources for consistent implementation, but some implementation is essential [25]. Each of these interventions has pros and cons (Figure 2).

### 4.1. What Interventions Are Effective Prior to or at the Time of Prescription?

#### 4.1.1. Core Strategy

Pre-authorization of restricted antimicrobials: A pharmacist or an ASP clinician must approve the prescription of a specific antimicrobial before the pharmacy releases it. This provides direct control over restricted antimicrobials and mainly improves the empiric use. The pitfalls of this strategy are the reduced autonomy of the prescribers and possible delays in the drug administration [16,26].

#### 4.1.2. Minor Elements

Local syndrome-specific clinical guidelines and pathways implementation: Local clinical guidelines (LCGs) or clinical pathways (CPs) targeting common syndromes (e.g., respiratory tract infections, urinary tract infections, skin infections, and surgical prophylaxis) are adapted from national or international evidence-based guidelines to fit specific healthcare settings, addressing local epidemiology, diagnostic capabilities, and drug availability [16,24]. According to recommendations from the World Health Organization (WHO) and the IDSA/SHEA regarding ASP implementation, local clinical guidelines should be accompanied by a strategy for implementation (i.e., healthcare worker education, clinical pathways creation, or audit and feedback [24,25]. Moreover, to guide clinical guideline contextualization, tools such as weighted-incidence syndrome combination antibiograms (WISCA) should be adopted. Indeed, WISCA provides highly informative estimates on antibiotic coverage patterns weighted on the most prevalent pathogens, overcoming the limitation of combination antibiograms [26,27,28].Healthcare worker education: Face-to-face or online talks, workshops, clinical case simulations, and toolkits are fundamental for optimal patient care to ensure knowledge of the most updated management strategies [29,30,31]. Continuous education will ensure that the most up-to-date treatments, diagnostic tools, and strategies are acknowledged. Healthcare worker education offers different advantages for stewardship training and can effectively improve prescribing behavior, especially when paired with other interventions [32].Computerized clinical decision support (CDS): Tools and mobile applications enhance ASPs by personalizing antibiotic regimens based on patient-specific factors and providing instant access to guidelines and dosing calculators, thereby improving adherence to stewardship principles and preventing dosing errors in real-time [33]. Like other ASP tools, CDS should be developed in alignment with guidelines established by national or international professional societies, supported by documented references.Caregivers education: Caregivers should be informed about the correct use, administration, storage, and disposal of antimicrobials, including antibiotics, to become allies in combating AMR and improve children’s outcomes. Mass education campaigns informing the public with messages, for example, about the ineffectiveness of antibiotics against viral infections such as respiratory syncytial virus (RSV) and influenza, or the importance of vaccines, can be implemented, as well as direct education from healthcare workers addressing specific syndromes. Messages should be communicated clearly and simply, considering the socio-cultural background, religious beliefs, and knowledge level of the parents being addressed. Leaflets in multiple languages could be distributed to parents to reach a wider audience and overcome language barriers. Including both approaches enhances overall public awareness of AMR and helps counter widespread misinformation and misconceptions about antibiotics. Interventions focusing on improving the quality of parent–healthcare provider communication have repeatedly exhibited the greatest impact on rates of inappropriate prescribing [34,35].Guidelines for antibiotic allergy and delabeling of spurious antibiotic allergies: Antibiotics are the cause of 40% of IgE and non-IgE-mediated allergic reactions to drugs, and the most allergenic are the beta-lactams. Still, up to 95% of patients reporting a penicillin allergy can tolerate a rechallenge [36,37]. Performing dedicated antibiotic allergy history-taking, with or without dedicated skin testing to remove false antibiotic allergy labels, can be an effective strategy to prevent the unnecessary avoidance of effective antibiotics and the indiscriminate use of broad-spectrum antibiotics [38].Antimicrobial order form: Generic antimicrobial order forms, in which the prescriber specifies the drug and regimen, may be utilized for any anti-infective or solely for restricted antimicrobials. These forms require the clinician to provide an indication for the antimicrobial and may also request the anticipated duration of therapy. By using these forms, documentation and communication regarding antimicrobial therapy can be enhanced, and data collection for medication use evaluations becomes more streamlined [39]. Electronic antimicrobial order forms should be preferred to written types because they reduce errors, standardize prescribing practices, provide decision support, enable better tracking and audits, and enhance overall efficiency and patient safety. When restricted agents are prescribed, these forms may require clinicians to confirm adherence to institution-specific criteria, supporting appropriate prescribing and simplifying the approval process.

### 4.2. What Interventions Are Effective After the Time of Prescription?

#### 4.2.1. Core Strategy

Prospective audits and feedback (PAF): Feedback can be provided in real-time or on a defined timing basis and can be delivered directly via the prescription tool or face-to-face during consultation meetings with the AST. This is usually a persuasive intervention where the rationale behind the recommendations is provided to convince the prescriber to modify the antimicrobial prescription but without imposing therapeutic choices (handshake stewardship). Different from the pre-authorization strategy, this intervention preserves the prescriber’s autonomy and allows for collaboration with the AST [40,41,42].

#### 4.2.2. Minor Elements

Dose optimization: Therapeutic drug monitoring (TDM) ensures that antibiotic levels remain within therapeutic ranges, optimizing efficacy while minimizing toxicity; regular monitoring and dose adjustments are particularly important for antibiotics with narrow therapeutic windows [43,44].Appropriate duration and antibiotic timing out: Ending antibiotic therapy after an appropriate length of treatment is crucial, as extending the duration unnecessarily can increase adverse events without improving patient outcomes. Moreover, setting specified timing-out of antibiotic refills will help reassess the antibiotic duration [16].From empiric to target therapy, based on culture results and antibiotic monitoring by pharmacist: Empiric therapy is started based on clinical judgment and likely pathogens, but once culture results are available, therapy should be tailored to the specific pathogens, determining the best possible combination of antibiotics with the help of a pharmacist, when needed, enhancing treatment efficacy and reducing unnecessary broad-spectrum antibiotic use [16,24].Switch to oral: Transitioning from intravenous to oral antibiotics when clinically appropriate (based on patient stability, ability to absorb oral medications, and availability of effective oral formulations) reduces hospital length of stay and healthcare costs [16].

By implementing a combination of these strategies, healthcare facilities can enhance their antimicrobial stewardship efforts, leading to better patient outcomes, reduced AMR, and overall improved public health.

## 5. Different Types of ASP: Does the Same Program Fit All Settings?

In pediatric settings, the implementation of ASPs varies across inpatients, outpatients, and PEDs, each with its unique challenges and strategies.

### 5.1. Inpatient Pediatric Care

In 2014, the Centers for Disease Control and Prevention (CDC) recommended that all hospitals implement ASPs and published the “Core Elements of Hospital ASPs”, which outlined the main features to ensure the success of an ASP [45]. A systematic review describing ASP implementation in hospitals in the USA and Europe showed that PAF, guidelines implementation, and more specific approaches based on laboratory testing and checklists were the most used interventions [46]. Indeed, a USA study suggested that a combination of PAF and pre-authorization could enable ASPs to maximize the strengths of each strategy [42], whereas a study conducted in the UK showed a great impact of behavioral interventions [47].

Different studies demonstrated the efficacy of ASPs in pediatric settings, both in high and low and middle-income countries (HICs and LMICs). The quasi-experimental study by Newland et al. demonstrated the effectiveness of PAF in reducing antibiotic use, showing a monthly reduction in overall antibiotic days of therapy (DOT) and length of therapy (LOT) by 7% and 8% per 1000 patient days, respectively (*p* = 0.045) [48]. The impact was even more pronounced for specific antibiotics—ceftriaxone/cefotaxime, vancomycin, ceftazidime, and meropenem—for which the DOT and LOT decreased by 17% and 18% per 1000 patient days, respectively (*p* < 0.001), compared to the results for hospitals without ASPs [48]. Similar results were obtained by Hersh et al. that compared nine hospitals with ASPs and 22 hospitals without ASPs, showing an average monthly decline of 5.7% in DOT and of 8.2% in DOT of specific antibiotics (vancomycin, carbapenems, linezolid) in the hospital with ASP [49]. Furthermore, an Italian study demonstrated the efficacy of an ASP intervention conducted through observation, education, audit, and feedback, as well as the provision of an electronic app to support antibiotic prescription based on local susceptibility data [49]. The study reported a significant decrease in antibiotic consumption (−452.49 DOT/1000 patient days, *p* < 0.001) after the introduction of a mobile app, with a clear inversion of the access-to-watch ratio (from 0.7 to 1.7), without an increase in length of hospital stay, admission to pediatric intensive care unit, and in-hospital mortality [50]. However, simpler interventions, such as the implementation of CPs, have also proven effective in reducing both antibiotic prescribing rates and the duration of antibiotic therapy. For instance, a study by Rossin et al. demonstrated a reduction in broad-spectrum antibiotic prescriptions (from 100.0% to 38.5%) and a shortened hospital stay (from 13.5 days to 7.0 days) for children admitted with lower respiratory tract infections (LRTIs) following the introduction of a specific CP for managing community-acquired pneumonia (CAP) [51].

Similar results were also obtained in studies conducted in LMICs [52,53]. The study published by Rahbarimanesh et al. showed a reduction in the use of meropenem and vancomycin in a children’s hospital in Pakistan after the introduction of a specific ASP [53], whereas the study published by Opondo et al. reported that in hospitals with ASP interventions, the risk of antibiotic prescriptions for non-bloody diarrhea was 70% lower than in hospital without ASP interventions [52].

The efficacy of ASPs has also been evaluated in special settings, such as pediatric and neonatal intensive care units or oncology units. The study published by Wattier et al., conducted in a pediatric oncology and stem cell transplantation service in the USA, showed a reduction in tobramycin and ciprofloxacin use after the update of internal guidelines on the management of fever in neutropenia [54]. Similar results were reported in the study of Haque et al., conducted in a pediatric intensive care unit in Pakistan, which showed a 64% reduction in antibiotic use after ASP implementation [55].

### 5.2. Primary Care Setting

In 2016, the CDC published the Core Elements of Outpatient Stewardship [45]. Despite these recommendations, the best methods for conducting ASPs in the outpatient setting are currently unknown, and the uptake of outpatient ASP has remained low [56,57]. Implementing ASPs in the outpatient setting involves more challenges than those noted in the inpatient setting, such as lack of funding, difficulties in identifying a clinician leader, obtaining antibiotic prescription data to identify high-impact targets, tracking process improvements and clinical outcomes, and sustaining improvements over time [58,59,60,61]. Additionally, antibiotic use is influenced by various factors, including parents’ beliefs and behaviors, their understanding of antibiotics, prior experiences, and adherence and disposal instructions [58]. In England, the largest reduction in antibiotic use was reported with structural-level interventions attributed to policy and commissioning interventions, such as primary care financial incentives [47].

Different studies showed the efficacy of ASP in outpatient settings, both in HICs and LMICs. An Italian study conducted at a regional level reported a significant impact of a multifaceted ASP, including guidelines, e-learning, and prescription reports, in improving the rate and quality of prescriptions in primary care settings [59]. The study reported a substantial reduction in the annual prescription rate per 100 patients (9.33 to 3.39; *p* = 0.009), with a reduction in prescription rates of amoxicillin–clavulanate (50.25 to 14.21; *p* = 0.001) and third-generation cephalosporins (28.43 to 5.43; *p* < 0.01) [59]. Similar results were obtained in a randomized controlled trial conducted with primary care pediatricians in the USA via a personalized audit and feedback intervention. The prescription of broad-spectrum antibiotics decreased from 26.8% to 14.3%, with an absolute difference of 12.5%, among primary care pediatricians in the intervention arm, compared with an absolute difference of only 5.8% in the control arm [59]. Considering specific infectious diseases, off-guideline prescriptions for children with CAP decreased from 15.7% to 4.2% in the intervention arm, compared to the control arm, in which the decrease was from 17.11% to 16.3% [60]. Unfortunately, the results of this trial were not sustainable over time. Indeed, twelve months after the start of the study, the PAF was stopped, revealing, in the six months after the end of the program, a new increase in antibiotic prescriptions. These results underlined the difficulties in sustaining results over time and the importance of intervention, especially education intervention, for the effectiveness of the program [61]. Similar results were obtained in a randomized controlled trial conducted in 25 township hospitals in China, randomly allocated to the intervention group (12 centers) and to the control group (13 centers) [24]. The antibiotic prescription rate decreased from 82% to 40% in the intervention group and from 75% to 70% in the control group, determining an adjusted absolute risk reduction in antibiotic prescribing of −29% (95%CI −42 to −16, *p* < 0.0002) [62].

### 5.3. Pediatric Emergency Departments

PEDs may be considered a hybrid setting, merging elements of both inpatient and outpatient care. They face unique challenges, including logistical and provider-level barriers, as well as inherent difficulties specific to the ED environment. Physicians in PEDs significantly influence prescribing practices for patients admitted to the ward and those discharged home. Antibiotic prescribing in this setting is challenging due to the high turnover of both patients and healthcare providers, coupled with the need for rapid decision making, often in the presence of diagnostic uncertainties [63,64,65].

A recent systematic review reported that ASP implementation in PEDs in the USA and Europe focused mainly on multiple interventions, such as clinical practice guidelines (CPG) or CPs combined with education, but also on single interventions and other types of ASPs, such as validated clinical prediction models for pneumonia [64]. Other potential strategies that could be implemented to improve antibiotic prescribing include utilizing the CDS tool and PAF, establishing follow-up procedures, implementing safety netting systems, and delivering comprehensive training and supervision [64]. The impact of CPs on the management of acute otitis media and pharyngitis in an Italian PED has been demonstrated in the study published by Donà et al., which reported a reduction in broad-spectrum antibiotic prescriptions after the implementation of CPs (from 53.2% to 32.4%, *p* < 0.001 for acute otitis media; from 46.6% to 6.6%, *p* < 0.001 for pharyngitis) [32]. The same results were obtained by implementing the same CPs in other Italian PEDs, with a statistically significant reduction in the use of broad-spectrum antibiotics by 29.5–55.2% and 80%, respectively, for acute otitis media and pharyngitis [27]. Nevertheless, as reported by Gerber in the outpatient settings [61], the results were not sustainable over time in all the centers involved. Indeed, only two centers combined the introduction of CPs with educational talks and recall lessons over time, whereas the third center did not associate recall educational lessons and was not able to sustain the results. This study highlighted the importance of education in the achievement and sustainability of results. Similar results were reported in another study conducted in the USA, showing an increase in the percentage of prescriptions with the recommended agent at the appropriate dose and duration from a mean of 32.7% to 52.4% after the introduction of an ASP characterized by a combination of tracking and reporting, education and expertise, and action for policy and practice and commitment [48].

## 6. Different Types of Diagnostic Stewardship

In pediatric settings, DSPs are tailored to the unique features of young patients, such as the need for non-invasive or micro-invasive techniques and the ability to perform tests on small samples, particularly capillary blood samples [66,67].

Various diagnostic tests, including biomarker tests, point-of-care tests (POCTs), and multiplex polymerase chain reaction (PCR) panels can be integrated into clinical practice [68,69].

Biomarker analysis, such as C-reactive protein (CRP) and procalcitonin (PCT) testing, is a valuable tool for distinguishing bacterial from viral infections and optimizing antibiotic prescriptions [70], although there are no predefined cut-offs for these biomarkers to distinguish between viral and bacterial infections. The study published by Barbieri et al. showed that the implementation of CP for the management of LRTIs, including the use of CRP and PCT with pre-defined cut-offs, reduced antibiotic prescriptions from 36.2% to 12.5% (*p* = 0.036) in patients hospitalized for bronchiolitis [71]. The use of PCT has also been studied to shorten the antibiotic length of therapy. A recent meta-analysis showed that PCT-guided antibiotic therapy was associated with a significantly shorter length of antibiotic therapy compared with that of the control group (weighted mean difference, WMD, −2.22 days; 95% CI, −3.41 to −1.03; *p* <0.001), with a significantly lower rate of adverse events compared with those in the control group (relative risk, RR, 0.25, 95% CI 0.11–0.58) [72]. Some studies have shown that PCT levels correlate with bacterial infections, particularly in neonatal and pediatric intensive care settings, enabling clinicians to make more informed decisions regarding the initiation or discontinuation of antibiotics. The study published by Milcent et al. in 2016 showed that PCT provides higher accuracy in identifying invasive bacterial infection (IBI) in neonates and infants less than 90 days of age compared to that of CRP; indeed, the AUC ROC curve for the detection of IBI for PCT was higher than that for CRP concentration (AUC, 0.91; 95% CI 0.83–0.99; vs. AUC, 0.77; 95% CI, 0.65–0.89; *p* = 0.002) [70]. Instead, no difference in accuracy was observed in the identification of severe bacterial infection between PCT and CRP [70]. The use of PCT in PICU to reduce antibiotic consumption was studied by Katz et al. [24]. Antibiotic days of therapy were not significantly different between the procalcitonin arm (median 6.6, interquartile range [IQR], 3.1–10.9) and the usual care arm (7.6, IQR 3–11.8), although more antimicrobial stewardship was displayed in the procalcitonin arm compared to the control arm, with a higher adherence to the algorithm-based antibiotic recommendations in the procalcitonin arm (70%) [73]. Despite the usefulness reported in some studies, other authors recommended against the use of PCT testing alone due to the great variability in specificity across different cut-offs [74].

POCTs are diagnostic tests conducted at the time and place of patient care. These tests provide quick results, aiding timely decision making, especially in outpatient and emergency settings [75,76,77]. One of the most common POCTs used in pediatric settings is the urinary dipstick, a valuable tool to identify children with a suspected urinary tract infection (UTI), avoiding antibiotic prescription in those without suspected UTI [76].

In children, CRP-POCT reduced antibiotic prescribing when CRP cut-off guidance was provided, with no difference in clinical recovery, resolution of symptoms, or hospital admissions compared to those resulting from usual care [75]. For respiratory tract infections, POCTs like rapid streptococcal antigen and influenza tests can significantly reduce unnecessary antibiotic prescriptions, although additional evidence is still needed regarding these tests in children. Indeed, streptococcal tests potentially reduce antibiotic use by 25% [78], especially when associated with educational lectures, as reported by the study which observed a decrease of more than 50% in antibiotic prescription for children with sore throat after the introduction of a rapid streptococcal test [78]. However, additional studies are needed regarding the use of rapid viral testing, since some studies did not observe changes in the antibiotic prescription rate in the group of children evaluated using rapid viral testing. For example, the study published by Thibeault et al. showed no difference in the antibiotic prescription rate in children with positive rapid test results for RSV compared to children with negative tests [79].

Multiplex PCR (mPCR) panels can simultaneously detect multiple pathogens (e.g., viruses, bacteria, fungi) from a single sample [80]. This tool is particularly useful in diagnosing respiratory and gastrointestinal infections in children, where symptoms of viral and bacterial infections often overlap. It is also beneficial in diagnosing conditions like meningoencephalitis and sepsis. Compared to standard methods, mPCR panels provide quick identification of the causative pathogen and can lead to targeted antimicrobial therapy, reducing the use of broad-spectrum antibiotics [80]. Children admitted with respiratory symptoms and tested using a syndromic panel were less likely to receive empiric antibiotics (odds ratio 0.45, *p* < 0.001; 95% CI 0.39–0.52) and had a shorter duration of empiric broad-spectrum antibiotics (6.4 vs. 32.9 h; *p* < 0.001) [81]. However, other studies failed to demonstrate this reduction. The study published by del Rosal et al., evaluating the use of the mPCR respiratory panel in children with suspected viral pneumonia, showed no statistically significant differences in total antibiotics consumption (83% vs. 86%) or antibiotics given for ≥72 h (58% vs. 66%) when compared to an historical cohort [68]. A reduction in antibiotic prescriptions at discharge was observed in the intervention group (41% vs. 72%, *p* = 0.001) [68]. These results are similar to those reported by Rao et al. in their randomized controlled trial, which indicated that the use of rapid respiratory pathogen testing did not change the antibiotic prescription rate compared to that observed for usual practice (RR 1.1; 95% CI 0.9–1.4) [82].

The use of mPCR has also been shown to be effective for other pediatric infectious diseases. In cases of acute diarrhea, the use of multiplex PCR panels reduced inappropriate antibiotic use from 42.9% to 25.8% [83], while the use of mPCR panels for central nervous system fluid analysis may reduce the duration of therapy and the hospital length of stay in children admitted for presumptive meningitis or encephalitis [84].

## 7. Different Types of Diagnostic Stewardship: Does the Same Program Fit All Settings?

### 7.1. Inpatient Pediatric Care

Hospitalized children often require immediate and effective antimicrobial therapy, making DSPs crucial in this setting. As part of hospital ASPs, molecular assays for rapid identification of pathogens in bloodstream infections (BSIs) have proven effective in minimizing the duration of empirical therapy and reducing the time to targeted antibiotic therapy in life-threatening neonatal and pediatric infections, encouraging de-escalation of antibiotics, when appropriate, and improving patient outcomes [67,85]. A study on the adult population reported a shorter median time until pathogen detection in the T2 group compared to the usual care (4.5 h vs. 60 h, *p* < 0.001) group, as well as the time until targeted therapy (antibiotic with the narrowest spectrum and maximal effectiveness, 6.4 h vs. 42.2 h, *p* = 0.043) [86]. Moreover, commercially available molecular methods for detecting antibiotic resistance genes within hours are promising tools to rapidly optimize treatment and aid in infection control by identifying clusters of resistant strains. Studies on the adult population showed encouraging results. A randomized controlled trial compared blood culture testing, using standard-of-care (SOC) culturing and antimicrobial susceptibility testing, with rapid organism identification and phenotypic antimicrobial susceptibility testing using the Accelerate Pheno system (RAPID) [87]. Although no difference in patients outcome between the two groups was observed, the median time to first antibiotic modification for overall antibiotics for Gram-negative antibiotics and for antibiotic escalation for antimicrobial-resistant BSIs was faster in the RAPID arm vs. the SOC arm (8.6, IQR 2.6–27.6 vs. 14.9, IQR 3.3–41.1 h, *p* = 0.02; 17.3, IQR 4.9–72 vs. 42.1, IQR 10.1–72 h, *p* < 0.001; 18.4, IQR 5.8–72 vs. 61.7, IQR 30.4–72 h, *p* = 0.01, respectively) [87]. However, considering their high costs, they should be reserved for specific settings, such as critically ill and immunocompromised patients.

For central nervous system infections, the FilmArray Meningitis/Encephalitis Panel can detect a broad range of pathogens directly in cerebrospinal fluid, with a turnaround time of about one hour [88], allowing for more targeted use of antimicrobials, particularly in young patients or specific populations, such as immunocompromised individuals. The study published by Kadambari et al., comparing a group of children that underwent lumbar puncture for suspected central nervous system infection before and after the introduction of FilmArray, showed a shorter duration of antibiotic use in the intervention group, especially in the case of enterovirus meningitis (median: 4 vs. 5 days) and human parechovirus meningitis (median: 4 vs. 4.5 days) but also in the case of culture/FilmArray-negative cerebrospinal fluid (median: 4 vs. 6 days) [89].

However, the risk of overtesting can lead to overdiagnosis, unnecessary antimicrobial treatments, and excessive costs. Inadequate specimen collection and storage can also reduce test accuracy. Therefore, robust microbiology support with pediatric-specific expertise, evidence-based CPs for diagnosing common pediatric infections, and educational programs for clinicians and nursing staff, alongside ASPs, are critical for obtaining accurate sample collection, timely processing, and proper interpretation of results to ensure appropriate diagnosis and achieve a consistent and cost-effective reduction in antibiotic consumption [90].

### 7.2. Primary Care Settings

In the primary care setting, POCTs are valuable tools beyond clinical symptoms to differentiate between viral and bacterial pathogens. Training pediatricians on the appropriate use of diagnostic tests, such as pharyngeal swabs or rapid viral tests, improves diagnosis accuracy, reduces unnecessary antibiotic use, and encourages “watch and wait” approaches before starting antibiotics. However, barriers exist to the widespread use of POCTs at primary healthcare levels, likely due to insufficient healthcare worker training or limited economic resources [91].

A recent meta-analysis showed that streptococcal rapid tests and influenza rapid tests are the most-used POCTs in primary care [92]. Many studies reported a significant reduction in antibiotic consumption following the introduction of the streptococcal rapid tests. Conversely, a significant increase in oseltamivir prescribing with the use of POCTs was observed in 60.0% of the analyzed studies. Implementing community-level education campaigns promoting accurate diagnosis and appropriate antibiotic use helps build public understanding and support for ASP initiatives.

Despite the costs related to the POCTs, similar good results were also achieved in developing countries. The use of a digital clinical decision support algorithm, combined with CRP testing, hemoglobin testing, a pulse oximeter, and mentorship has proven effective in reducing antibiotic prescription in primary care facilities in Tanzania, with an adjusted difference of −46.4% (95% CI −57.6 to −35.2), without an increase in treatment failure (adjuster relative risk 0.97, 95% CI 0.85–1.10) [93]. Another study conducted among primary health facilities in Burkina Faso evaluated the impact of POCTs for respiratory pathogens (RSV, influenza, *Streptococcus pyogenes*, *Streptococcus pneumoniae*), for malaria, and for other infections (dengue and typhoid fever), showing a reduction in antibiotic prescription (risk difference (RD) −16.8%, 95%CI −21.7% to −12.0%, *p* < 0.010), with a greater decrease in patients without malaria (RD: −46.0%; −54.7% to −37.4%; *p* < 0.001) and in those with a respiratory diagnosis (RD: −38.2%; −43.8% to −32.6%; *p* < 0.001) [94].

### 7.3. Pediatric Emergency Departments (PEDs)

The PED is a dynamic environment in which rapid decision making is essential. Antimicrobial treatment is typically empiric, often without microbiological results or feedback on the patient’s course. The use of POCTs has proven effective for ASPs in PEDs, significantly decreasing antibiotic use after their introduction into clinical practice [77]. Indeed, the study published by Tan et al. showed that a positive rapid viral test was associated with fewer antibiotic prescriptions compared the results when no test was performed (aOR 0.6, 95% CI 0.5–0.9.) [77]. However, the implementation of these tests alone is often insufficient to modify antimicrobial prescription trends. The introduction of POCTs should be accompanied by educational interventions and the establishment of diagnostic and therapeutic CPs to ensure the rational use of antibiotics [77].

## 8. How Should the Effectiveness of an ASP and DSP Be Evaluated? The Different Metrics in Pediatric Settings

Measurement performance indicators are essential to evaluate the effectiveness of ASPs. Standardized measurements are challenging to assess and vary among studies in different settings. Most of the studies evaluated antibiotic use [1]. The DOTs per 1000 patient days is preferred over defined daily doses (DDDs) per 1000 patient days in the hospitalized pediatric population because the DDD does not accurately describe antimicrobial use in pediatric settings due to age- and weight-based dosing in children [95]. DOT is defined by counting each antimicrobial administered on a hospital day (i.e., a 5-day course of three antibiotics results in 15 DOTs) [25].

The LOT per 1000 patient days provides an overview of the number of days patients receive antimicrobial therapy, irrespective of the number of antibiotics prescribed, and could be an additional metric to describe overall antimicrobial prescribing behaviors in hospital settings [96]. The DOT/LOT ratio measures the number of antibiotics prescribed to each patient. Notably, unlike the DDD per 1000 patient days, DOTs and LOT can only be obtained via medical records [96].

These metrics fail to describe prescribing appropriateness, including antibiotic dose accuracy, the spectrum of activity (i.e., a broad-spectrum antibiotic like a carbapenem usually counts for half the DOTs compared to a narrow-spectrum combination course), and reasons for prescribing [25,96]. To better characterize antibiotic prescriptions, the WHO developed the AWaRe classification [24]. This is a valuable framework for antibiotic stewardship, as it simplifies prescribing by categorizing antibiotics into Access, Watch, and Reserve groups, based on resistance potential and clinical necessity. “Access” antibiotics are those antibiotics that could be prescribed freely and should always be available, such as amoxicillin, amoxicillin–clavulanate, and first-generation cephalosporins; “Watch” antibiotics are those that should be prescribed with caution due to a broader spectrum of activity and a higher risk of developing antibiotic resistance, such as second- and third-generation cephalosporins, macrolides, carbapenems, and fluoroquinolones; “Reserve” antibiotics are 29 newer-generation and more expensive antibiotics, such as linezolid, daptomycin, ceftazidime–avibactam, or meropenem vaborbactam, which are the last option for multidrug-resistant (MDR) infections [97,98]. A unique feature of the system is its inclusion of unclassified antibiotics, which often consist of fixed combinations and serve as a point of interest for developing country-specific stewardship strategies. This unclassified group spans both narrower-spectrum and broader-spectrum antibiotics, reflecting varied use patterns that underscore the need for clearer guidance regarding their categorization. In countries with high use of unclassified antibiotics, reclassifying these agents under AWaRe categories may shift access-to-watch ratios, potentially complicating trends and stewardship approaches for these medications. Although tools like the access-to-watch ratio and the amoxicillin index provide intuitive methods for evaluating antibiotic use, they have limits. For instance, these tools may overlook narrower-spectrum antibiotics in certain regions, and they lack the ability to directly address the antibiotic spectrum, highlighting an area where AWaRe could be refined to support more detailed stewardship efforts [99].

To better assess prescribing appropriateness, antimicrobial use measures should be adjusted by case mix and ward [100]. Other qualitative indicators applied should evaluate adherence to guidelines (i.e., the proportion of children on antibiotic therapy without indication; surgical prophylaxis beyond 24 h; time to switch from empiric to targeted antibiotic treatment when etiology has been assessed, or from intravenous to oral treatment, when indicated).

Gathering antibiotic data is essential for evaluating the effectiveness of a stewardship intervention. Although daily data collection is ideal, it is often labor-intensive and consumes significant human resources, especially when electronic data are not available. Point prevalence surveys (PPSs) have become a valuable tool for efficiently capturing antibiotic prescription information with reduced effort [101].

In the outpatient pediatric setting, the ASPs’ most used indicator is the number of antibiotic prescriptions per 1000 children per year [102]. More useful metrics include the prescription rate for Watch antibiotics classes and the amoxicillin/amoxicillin–clavulanic acid ratio. Amoxicillin currently remains the first-line antibiotic recommended worldwide for most of the pediatric respiratory infections, accounting for more than 70% of pediatric visits. Therefore, an increase in the amoxicillin to amoxicillin–clavulanic acid ratio is a marker of effective pediatric ASPs in the community.

Microbiological outcomes, such as the rate of MRSA and ESBL or carbapenem-resistant *Enterobacterales* in invasive infections, are often proposed. In particular, the UKHSA proposes AMR testing every three months as local indicators to monitor *Escherichia coli* resistance to 3rd generation cephalosporins, carbapenem, ciprofloxacin, gentamicin, and piperacillin/tazobactam in blood samples and to trimethoprim and nitrofurantoin in urine samples [103]. Moreover, it is recommended to produce indicators for each hospital and for each local health unit. Indeed, different studies showed a reduction in AMR after the introduction of ASPs. For example, in the study published by Sarma et al. in 2015, after the introduction of fluoroquinolone restrictions, a decline in the percentage of ciprofloxacin-resistant extended-spectrum beta-lactamase (ESBL) producing urinary *E. coli* isolates was observed in both hospitals (RR: 0.473; 95% CI 0.315–0.712) and community settings (0.098; 95% CI 0.062–0.157) [104]. Another study published by Lawes et al., focusing on the restriction of the use of cephalosporins, co-amoxiclav, clindamycin, fluoroquinolones, and macrolides, showed a decrease in the methicillin-resistant *Staphylococcus aureus* (MRSA) prevalence density by 54% in hospitals and 37% in the community [105]. Other studies reported a decline in macrolide-resistance rates in Gram-positive cocci following a reduction in macrolide prescriptions in the community [106,107]. However, a lack of decreasing prevalence of MDR pathogens may not reflect ASP failure. Indeed, AMR is a complex event that probably requires time to reverse after a decrease in antimicrobial pressure.

The rate of *Clostridioides difficile* infections (CDIs) as an ASP measure should be used cautiously in the pediatric setting due to the high carriage rate in the first years of life, different from rates in the adult setting [108]. However, the hospital-acquired CDIs rate should be monitored, particularly among immunocompromised patients beyond two years of age, as it is a good marker of poor strategies for containing healthcare-associated infections other than through the use of antimicrobial prescriptions [109]. Again, the UKHSA recommends producing CDIs rates every twelve months for each hospital and for each local health unit for both community-acquired and hospital-acquired infections [103]. Moreover, other indicators recommended for monitoring inlcude hospital and community bacteremia rates caused by *E. coli*, *Klebsiella* spp., MRSA, MSSA, and *Pseudomonas aeruginosa*.

Clinical outcome measures are more difficult to obtain in children. Mortality is very rare in children and, like intensive care admission, could be a multifactorial event often associated with underlying clinical conditions. Metrics that are considered reliable and feasible in pediatric hospital settings include the length of hospital stay (LOS) and the 30-day readmission rate. Recently, the 30-day clinical failure rate has been more frequently reported to evaluate clinical outcomes, particularly in studies assessing shortened antibiotic courses for subacute (such as osteomyelitis) or non-invasive infections [60].

## 9. What Is the Cost-Effectiveness of an ASP and a DSP?

Different ASP and DSP strategies can directly and indirectly affect healthcare expenses, and the planning and implementation of each intervention incur costs. Therefore, the cost–benefit balance of ASPs and DSPs should be considered in experimental models testing different approaches. The models of economic evaluation include the following [110]:Cost-minimization analysis: Two alternative programs or treatments are compared to ascertain the least expensive. This type of analysis is helpful to ascertain the short-term impact of an ASP, while it does not address the long-term impact, especially AMR;Cost–benefit analysis: The costs associated with an ASP strategy should be compared to the potential benefits, including both financial metrics and intangible benefits that encompass health gains, lives saved, and the reduction in adverse events following antimicrobial use (e.g., CDIs);Cost-effectiveness analysis: The cost difference between two interventions (or compared to standard care) divided by the difference in their effects, is defined as the incremental cost-effectiveness ratio (ICER). It represents the average incremental cost associated with one additional unit of effect (e.g., the incremental cost per percentage point reduction in antibiotic prescription rate). This is the simplest measure of economic analysis, although it may not account for confounding factors such as hospital occupancy rates, price variations, and cost differences among different producers. This type of analysis should be normalized using consumption metrics such as DDD, DOT, and LOT [110].

The most straightforward approach is based on the comparison between the costs of antimicrobial pre-intervention and post-intervention. Indeed, most studies published in pediatric settings focus on antibiotic cost savings before and after an ASP implementation, typically reporting only short-term evaluations. In the study published by Zhang et al., the implementation of a multifaceted ASP in the outpatient setting has been associated with a 29% absolute risk reduction in antibiotic prescribing in children with upper respiratory tract infection, with an incremental cost-effectiveness ratio of only USD 0.03 per percentage point reduction in antibiotic prescribing [111]. Another study, published by Velasco-Arnaiz, reported that the hospital expenditure on antibacterials and antifungal drugs decreased by a total of EUR 64,406 in 2017 and EUR 137,574 in 2018 as compared with the 2015–2016 mean expenditure, for an absolute savings of EUR 201,980 after the introduction of a post prescription review with a feedback-based antimicrobial stewardship program in a European children’s hospital [41]. However, costs extend beyond the direct expenses associated with the antimicrobials themselves and should also include various fixed or potential costs, such as hospital length of stay, resource utilization, mortality rates, and rehospitalization rates, which are more challenging to quantify.

Cost-analysis can vary in complexity, even for the same variable. For instance, the direct costs of antimicrobial treatment can be categorized into three levels: the first level includes only drug acquisition costs; the second level incorporates the first level along with costs related to preparation, dispensing, and administration, as well as expenses related to antibiotic-related adverse events and clinical failure; and the third level further adds daily hospital stay costs [112,113].

The reduction in costs after the implementation of DSPs is still debated. The study published by Lubell et al. regarding the use of CRP POCT in the management of respiratory tract infection in children in primary care settings in Vietnam did not show a reduction in healthcare costs. Indeed, although the use of this POCT reduced the consumption of antibiotics, the cost of the test itself was higher than the potential savings from reduced prescriptions observed in the study. However, the authors reported that with higher adherence to the test results, their use would be cost-beneficial [114].

Indirect costs are frequently overlooked, as are long-term effects. For example, cost savings related to the reduction in AMR are not frequently estimated. Furthermore, there is a high variability in costs attributed to the same intervention and to AMR (if estimated) in both inpatient and outpatient settings, which could influence the analysis. A systematic review and meta-analysis published in 2023 estimated that infections caused by MDR bacteria, in comparison with non-MDR types, exhibit higher management costs, estimated up to USD 29,000 per episode [115]. This higher cost could be related to the higher length of stay and the higher odds ratios for resistant infection and mortality [115]. Table 1 reports a detailed list of costs and potential benefits of ASPs, according to the healthcare setting [116].

Moreover, when evaluating cost-effectiveness, the type of healthcare systems should be considered since in some nations, healthcare costs are entirely covered by patients, creating potential barriers to access and treatment, while in others, they are fully or partially covered by the state, ensuring more equitable access to medical services. These differences profoundly influence healthcare delivery, patient behavior, and implementation of public health policies.

Additional elements that impact economic outcomes include the type of healthcare system financing and the care setting in which the program is implemented. For instance, in hospital settings, costs primarily involve the work time of healthcare professionals and the implementation of diagnostic techniques, while benefits can be easily estimated by comparing expenditures on antimicrobial acquisition before and after the intervention. In the study published by Turner et al., the implementation of an ASP consisting of physician-group engagement and pharmacist PAF in a freestanding children’s hospital in the USA showed a reduction in the average monthly drug-acquisition costs [117]. Although pharmacy and physician clinicians were asked to perform additional tasks for the ASP, these tasks were integrated into the clinical practice, resulting in no additional cost. Therefore, the study showed a cost saving of approximately USD 67,000/year over the 2-year post-intervention period [117]. Conversely, in outpatient settings, the costs are often incurred by general practitioners or primary care providers. Furthermore, regional variations in healthcare systems (such as public financing, insurance-based, or out-of-pocket models) also contribute to differences in economic outcomes. Table 2 summarizes a framework for establishing costs accountable for AMR in a one-health perspective.

## 10. Which Are the Most Important Phases to Successfully Implement an ASP?

Implementing an ASP requires careful adaptation to specific settings, also considering hospital capacity, patient complexity, and hospital workload, as no fixed model or consensus regarding the best approach exists [24,118,119]. The process can be divided into three phases, as shown in Figure 3.

### 10.1. Planning Phase

This is a critical phase for establishing a robust program. To achieve more precise interventions, it is essential to focus on specific pediatric infectious diseases, such as CAP or UTIs, rather than on broad objectives, like reducing unnecessary vancomycin use [118,119]. Specific professional competencies in ASPs and infection management are needed, and the interventions chosen should align with your goals, the literature search, and available resources and settings. Some interventions are costly and may not be feasible everywhere. For example, a PAF intervention might be challenging in a PED but more suitable in an intensive care unit with adequate resources. Moreover, compliance with national and local guidelines is essential unless strong evidence supports alternative approaches. The creation of teamwork and the definition of the roles are essential steps. Furthermore, it is important to pre-establish the outcomes that should be evaluated, identify the best data collection methods, and evaluate whether all the data needed for the analysis are available. This phase, as well as all the other phases, can be costly, both financially and in terms of human resources, so finding funds and dedicated personnel is mandatory [118,119].

### 10.2. Implementation Phase

This phase involves executing the planned strategies. To better engage local staff, it is essential to identify a local leader to assist with implementation and to serve as the local reference point. This leader should motivate team members, foster a positive work environment, and facilitate communication among healthcare personnel [120].

Moreover, face-to-face meetings with healthcare staff and the hospital board are crucial for fostering a collaborative environment and securing funding. During these meetings, it is essential to clearly explain the project and its aims.

Monitoring the implementation process through scheduled meetings is essential to assess difficulties encountered and to allow for adjustments to the program, if needed. Moreover, it allows for the sharing of interim results, the demonstration of achievements, and the encouragement of continuous improvements. Indeed, real-time feedback reinforces efforts and allows for promptly addressing any setbacks [118,119].

### 10.3. Monitoring and Sustainability Phase

The final phase evaluates the results and ensures the ASP’s long-term success. Collecting data for monitoring, whether manually, electronically, or through PPSs, ensures the collection of comprehensive information for a thorough analysis, which should be conducted using appropriate metrics [118,119]. To maintain transparency and engagement, the results should be presented to stakeholders through face-to-face meetings, including comparisons with other settings. Furthermore, it is important to be receptive to criticism and challenges to adjust aspects of the ASP to better suit the specific setting. Attention to both positive and negative feedback and assessing challenges during implementation help to enhance the ASP [118,119].

In this phase, it is also important to collaborate with the team and stakeholders to determine the most effective methods for sustaining and improving the achieved results [71].

## 11. Future Perspectives

Core strategies such as including prospective audits, feedback, and pre-authorization have shown effectiveness in ASPs but are often resource-intensive [121,122,123]. Unlike in inpatient settings, in pediatric outpatient settings—where most antibiotic prescriptions occur [123]—stewardship efforts face additional challenges due to limited resources, lack of structured frameworks, and insufficient organizational support [124]. As a result, interventions are often guideline-based, while more comprehensive strategies are rarely implemented. To address these challenges, key strategies have been highlighted. The first is commitment, urging prescribers to engage actively in responsible antibiotic use. Next, action for policy and practice focuses on adopting specific policies or interventions aimed at improving antibiotic prescribing. Tracking and reporting involve the continuous monitoring of prescriptions, coupled with feedback through audits, to help prescribers assess and refine their practices. Lastly, education and expertise should extend beyond healthcare providers to include patients and caregivers, ensuring a comprehensive approach to improving antibiotic use and understanding [125,126]. Parents play a critical role in pediatric ASPs, as they are often the primary influencers in their child’s healthcare decisions. Their expectations and knowledge about antibiotics can significantly affect prescribing behaviors, either promoting or discouraging the use of antibiotics [58]. Educating parents about the appropriate use of antibiotics and the risks associated with resistance is crucial to minimizing unnecessary prescriptions. Despite its significance, the involvement of parents in stewardship efforts remains underexplored. Further research is needed to identify effective strategies for parental engagement, evaluate the impact of educational interventions, and develop methods to integrate parents as active partners in stewardship. Understanding these elements is essential for creating comprehensive ASPs that address both the provider and parental factors influencing antibiotic use in pediatric care.

Additionally, POCTs, mPCR, septic biomarkers, and reliable microbiological cultures play a critical role in timely treatment and reducing unnecessary antibiotic use in both pediatric outpatient and inpatient care [127]. However, assessing the financial and logistical aspects of implementing POTCs in pediatric care is essential to gauge their long-term viability and impact within healthcare systems. Key considerations in implementing POCTs extend beyond the initial costs and logistical planning. Proper training for healthcare professionals is crucial not only to use POCTs effectively but also to interpret the results accurately, as misinterpretation could lead to inappropriate treatment decisions. Ensuring widespread availability across diverse clinical settings, particularly in resource-limited areas, requires coordinated infrastructure and sustained funding. Integrating POCTs within existing healthcare workflows further supports their effectiveness, maximizing their potential impact on patient outcomes and ASP effort [127]. Further research is needed to evaluate how these diagnostic tools can be seamlessly incorporated into daily clinical workflows and to optimize their use for sustainable, effective pediatric care

Furthermore, in specialized pediatric fields such as neonatal or pediatric intensive care units and hematology–oncology wards, the complexity of care is compounded by the need to manage infections caused by MDR organisms. While novel antimicrobials are more readily available for adult populations, children often face limited access due to ethical concerns surrounding drug trials in pediatric patients, as well as the complexities involved in conducting such trials [128]. The lack of specific pediatric data can result in suboptimal dosing strategies, as the pharmacokinetics and pharmacodynamics of drugs in children differ significantly from those in adults, necessitating tailored approaches. On the other hand, the risk of irreversible harm, such as profound ototoxicity associated with aminoglycoside use in neonates, has made these antibiotics challenging to utilize. However, the ability to rapidly identify common genetic mutations predisposing patients to this adverse effect through POCT could significantly impact their use in NICUs [129]. This is particularly important, as aminoglycosides play a critical role in reducing reliance on carbapenems Moreover, in these settings, critically ill children are especially vulnerable, making it difficult to distinguish infections from other inflammatory conditions. This ambiguity complicates decisions regarding de-escalating or discontinuing antibiotics, leading to higher antibiotic usage compared to that in general pediatric wards [130]. Implementing targeted interventions to refine empiric antibiotic therapies can reduce inappropriate broad-spectrum antibiotic use. Updating internal guidelines with the latest evidence and applying the WISCA method to identify optimal empiric treatments are promising strategies to improve antibiotic prescribing for these fragile patients [28].

One critical area in the management of these complex cases is TDM. It is essential for ensuring that the dosage of antimicrobials is optimized to maximize efficacy while minimizing toxicity [131]. Pediatric patients, especially neonates and those who are critically ill, often require precise dosing adjustments due to factors such as varying body sizes, organ immaturity, and complex underlying health conditions that influence drug absorption, metabolism, and excretion. Additionally, many antimicrobials used to treat MDR organisms have a narrow therapeutic window, underscoring the importance of TDM. TDM is essential for preventing underdosing, which can result in treatment failure, and overdosing, which may lead to severe adverse effects, thereby ensuring optimal therapeutic outcomes and patient safety. TDM-guided expert clinical pharmacological advice programs have proven effective in tailoring and optimizing treatment dosage, resulting in more than 40% of the recommended dosing adjustments in both adult and pediatric setting [132].

Establishing ASPs in LMICs involves a distinct set of challenges. Infectious diseases continue to be the leading cause of mortality among children under the age of five, and AMR rates in these regions are notably high [133]. Restricted access to microbiology services and antibiotic susceptibility testing frequently leads to extensive reliance on broad-spectrum antibiotics [134]. Additionally, healthcare resources in developing countries are frequently limited, with professionals who may not always have specialized training. Guidelines targeted for establishing ASPs in LMICs are essential and should focus on different areas compared to those for HICs, such as, for example, ensuring access to diagnostic testing; providing education about AMR and antibiotic prescription practice; and establishing national and international agencies to regulate, monitor, and audit drug production, distribution, and dispensing practices [134].

Data collection is crucial, yet challenging. Electronic daily data collection provides a clear, representative view of antibiotic usage and highlights changes in prescription patterns post-intervention. However, since many centers lack electronic data systems, manual collection is often required. To lessen the burden on healthcare staff, PPS, conducted at various times throughout the year, offers a practical alternative for gathering antibiotic prescription data efficiently [135].

The use of varied metrics and the absence of internationally validated measures for pediatric patients complicate comparisons across similar studies in different settings. While DDDs have been applied in pediatric research, uncertainty about their suitability remains. The AWaRe classification is a step forward toward standardized metrics in pediatric settings; however, the large number of antibiotics in the unclassified category poses challenges, particularly in countries with high usage rates for these antibiotics [9].

One major challenge in implementing pediatric ASPs is the need to revise regulatory frameworks and develop specific recommendations that account for global healthcare diversity. Existing guidelines, largely designed for the USA healthcare system [16], must be adapted to suit the varying structures in different settings in other countries. National ASPs have already been established in some countries outside the USA. For example, since 2013, the UK has developed and adopted a variety of interventions to improve ASPs, with the aim of reducing antibiotic consumption, especially broad-spectrum types, in both primary and secondary care [136].

Although ASPs consistently enhance clinical practices and patient outcomes, their return on investment is often indirect and not immediately visible [136]. Developing a comprehensive business plan is crucial for the success of an ASP; however, integrating stewardship responsibilities into the roles of existing staff can serve as a more cost-effective strategy. This approach maximizes the use of current resources, while still supporting the goals of the program, allowing institutions to implement ASPs without incurring significant additional costs. However, for long-term success, developing a comprehensive business plan that ensures sustainable financial backing for both personnel and infrastructure is essential.

Despite considerable advancements in pediatric ASPs over the past decade, significant knowledge gaps remain in pediatric care, highlighting the need for ongoing research to refine and enhance these programs. Future efforts must prioritize large-scale, longitudinal studies to evaluate the long-term impact of ASPs on AMR trends, clinical outcomes, and overall patient safety. These studies should incorporate standardized metrics, allowing for comparability across diverse healthcare settings and regions. Such a comprehensive approach would enable the identification of global patterns in ASP effectiveness, helping to establish benchmarks for success and revealing context-specific challenges that require targeted interventions.

Investigating the economic impact of ASPs, particularly in resource-limited settings, is crucial for understanding how these initiatives can be sustainably scaled and maintained. Such data would provide valuable insights to guide policy decisions, advocating for increased investment in healthcare infrastructure and training to support pediatric ASPs.

Overcoming financial and infrastructural barriers through innovative approaches like telehealth and artificial intelligence (AI) is essential for extending the reach and effectiveness of ASPs globally, ensuring their successful implementation across diverse healthcare environments [3].

The potential applications of AI in medicine are vast. Importantly, AI solutions are designed to complement, not replace, physicians and healthcare professionals, serving as valuable tools to improve and support clinical practice [3].

AI and machine learning (ML) could be useful in monitoring antibiotic prescribing. Indeed, antimicrobial prescribing requires frequent adjustments as clinical data evolve, but manual review is challenging due to limited resources and extensive information. Furthermore, implementing ML models that leverage variables from electronic medical records could accurately identify antimicrobial exposure, enabling meaningful comparisons of antibiotic use across different hospitals [137].

AI could play a pivotal role in the fight against AMR. ML algorithms could predict trends in AMR and treatment responses, suggest specific therapeutic agents, and identify genetic markers related to antibiotic resistance by analyzing large genomic datasets [138]. For example, Feretzakis et al. developed an ML-based model using antimicrobial susceptibility data from a tertiary hospital in Greece. This model predicts the resistance pattern of an isolate based on sample source, infection site, Gram stain results, and past susceptibility data, achieving 72.6% accuracy [139].

Hospitals increasingly use automated decision support systems, although many rely on static rule sets that generate excessive, unhelpful alerts. To address this, ML-based systems incorporating feedback to continuously improve and effectively support key decisions may be helpful [140].

AI could also be applied in drug development to streamline the discovery of new antibiotics and to optimize drug administration [141], as well as in expanding knowledge, particularly among students and residents, by enhancing the understanding of antibiotic prescribing practices [142]. However, a gap remains between promising AI models and their integration into clinical settings. Further research is needed to assess the impact of AI and ML on ASP, as well as to fully explore the capabilities these technologies may offer [143].

On the other hand, telehealth can help address staffing shortages in remote or underserved areas by enabling healthcare providers to offer timely guidance and support, thereby bridging critical gaps in care [144]. Access to technology is also helpful for delivering real-time, localized data on antibiotic use, with tools that facilitate detailed analysis and provide direct feedback, including alerts for broad-spectrum antibiotic use, automatic stop orders, and the transition from intravenous to oral therapies [144]. Integrating telehealth and AI into pediatric care holds promise for optimizing antibiotic prescribing, enhancing adherence to clinical guidelines, and supporting timely adjustments in therapy, including de-escalation.

Addressing these priorities will be key to advancing more effective and sustainable ASPs, ultimately reducing the global burden of AMR and improving health outcomes in pediatric populations.

## 12. Conclusions

This narrative review explores the key features and challenges associated with ASPs and DSPs in pediatric settings, emphasizing the need for tailored educational programs and cross-disciplinary collaboration, as well as laying the groundwork for the development of countries’ specific pediatric consensus guidelines.

A key component of each ASP and DSP is the evaluation of the effectiveness of the interventions through standardized metrics. However, in pediatric settings, these metrics are not yet well-established for either inpatient or outpatient care. More robust studies are needed to identify the most appropriate metrics tailored to pediatric populations. Additionally, future research should explore the integration of AI into ASPs and DSPs to enhance their effectiveness in combating antimicrobial resistance (AMR).

## Figures and Tables

**Figure 1 antibiotics-14-00132-f001:**
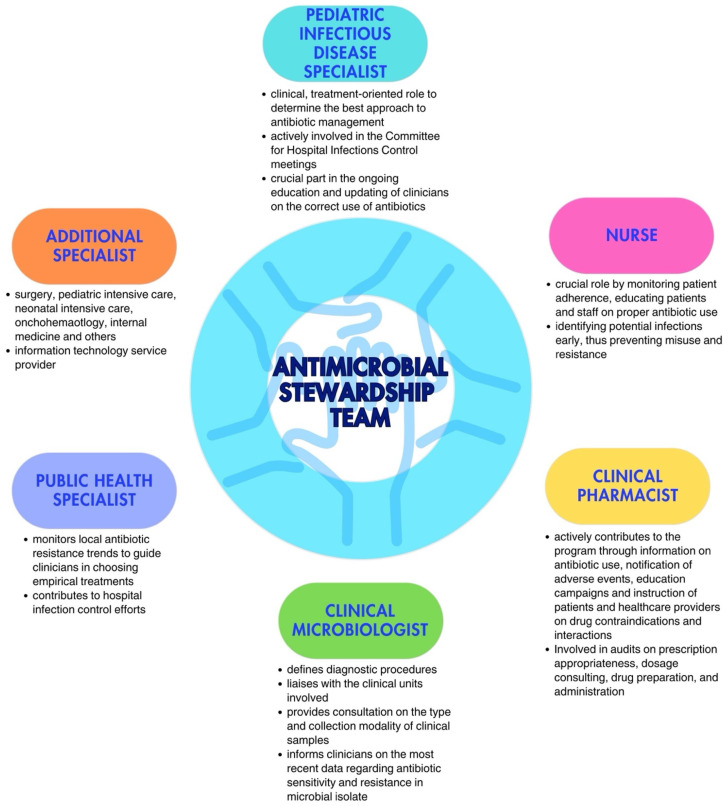
Key figures in the Antimicrobial Stewardship Team (adapted from Dellit et al. [16] and Donà et al. [20]).

**Figure 2 antibiotics-14-00132-f002:**
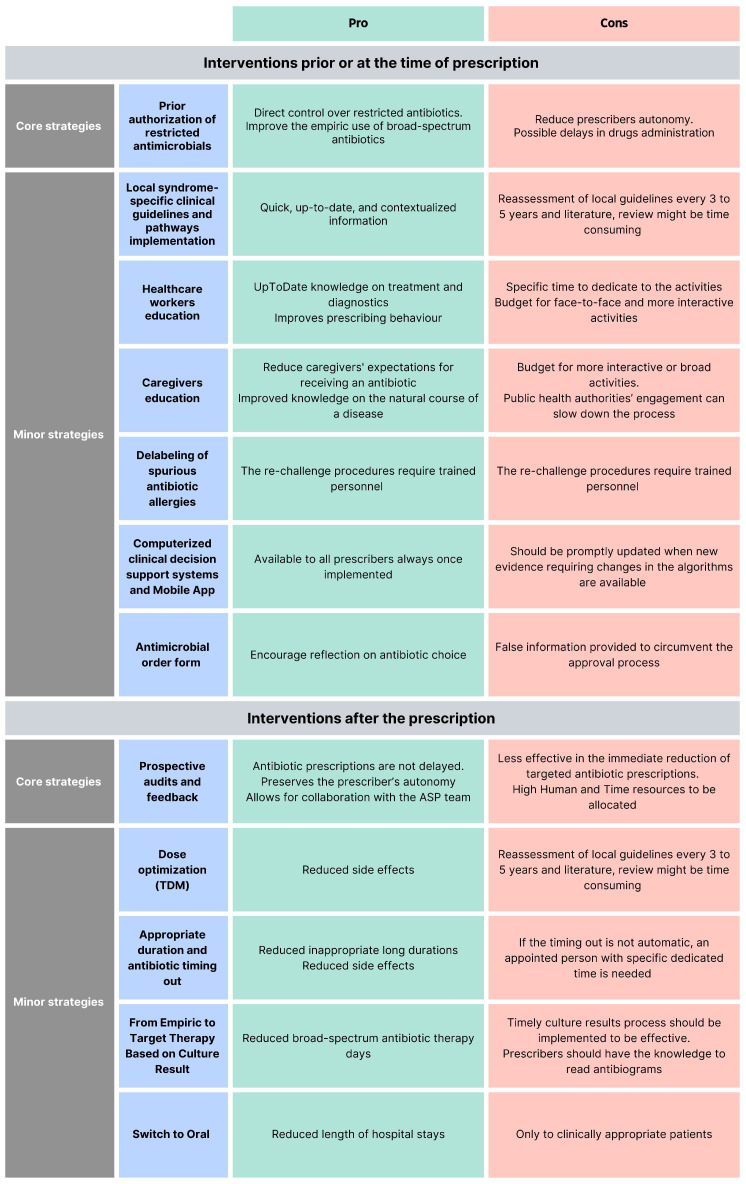
Pro and cons of different antimicrobial stewardship programs.

**Figure 3 antibiotics-14-00132-f003:**
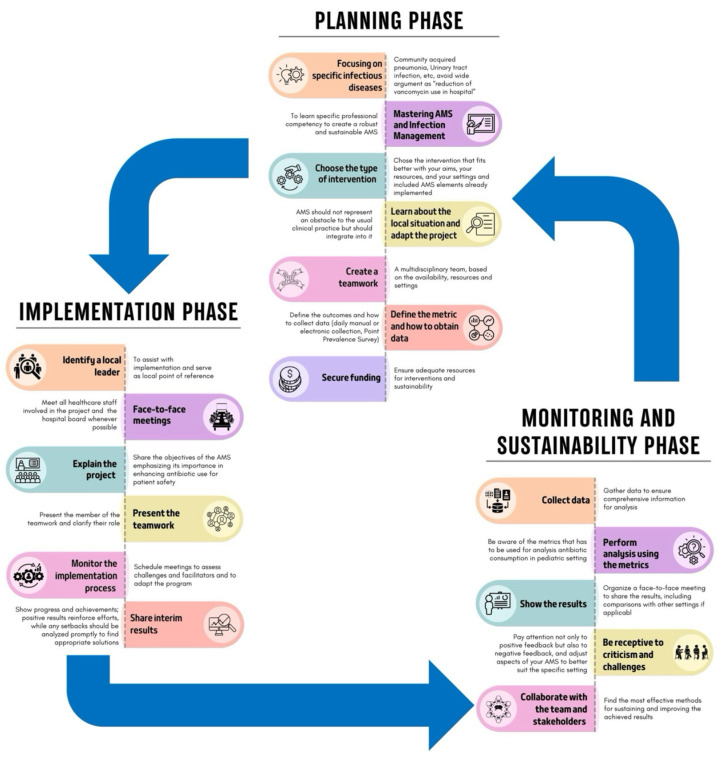
Phases of the implementation of an ASP (adapted from Mendelson et al. [119]).

**Table 1 antibiotics-14-00132-t001:** Incremental costs and potential benefits of antimicrobial stewardship programs.

Setting	Incremental Cost	Potential Saving/Benefit
Inpatient	Clinical pathways implementation: Molecular diagnostic (ex. multiple PCR assay, genotypic AMR characterization); POC-CRP or other POC tests (POC-NAAT, multiple biomarkers tests); Additional diagnostic tools. Implementation (operational) cost: Salaries for AMS personnel and data monitoring; Staff salaries and benefits; Computers and software; Training sessions; Circulars/educational materials; Therapy evaluation, review, and feedback.	Direct costs: Hospital costs per day/per bed Cost of patient isolation (supplies, housekeeping, waste disposal, increased portable testing services, and increased staffing) Physician staff time Costs related to antimicrobial therapies (acquisition and administration costs, specialized nurses staff time, catheter placing and medications for parenteral antimicrobials) Infection control staff salaries Adverse events or side effects by antimicrobials (medications and laboratory costs for screening) Costs related to the management of patients affected by AMR pathogens Indirect costs: Loss of productivity/earnings by patient, family and visitors during hospitalization
Outpatient	Medical consultations and revisits; Nursing support and data monitoring; Clinical pathways implementation: CRP-POC or other tests (NAAT, multiple biomarkers tests); Rapid tests (ex. GAS, SARS-CoV-2, etc.); Additional diagnostic tools/instruments. Training for healthcare professionals: Communication skills training for healthcare professionals; Educational leaflets; Clinical practice guidelines development and implementation.	Medical consultation for antimicrobial prescription; Costs related to antimicrobial therapies (reimbursement by the payer); Prescription monitoring or review; Ancillary tests (in emergency department); Medication costs.

**Table 2 antibiotics-14-00132-t002:** Human health costs related to the patient colonized or infected with resistant pathogens.

Direct Costs:	Indirect Costs:	Associated Probabilities:
Costs of any treatment or prophylaxis: Cost of antibiotics acquisition and administration (central lines, etc.), de-colonization (e.g., mupirocin); Non-standard surgical prophylaxis in colonized/infected patients; Cost of nursing care; Extended length of stay (+ cost of cohorting); Re-testing; Infection prevention and control interventions (e.g., screening at admission or before surgery). Costs of long-term consequences of AMR infection: Additional laboratory tests or imaging; Adverse events to 2nd and 3rd-line treatments, etc.; Monitoring of toxicity and efficacy; Hospital admissions, rehabilitation and/or treatments required for MDR infection sequelae. Out-of-pocket expenditure by the patient for care: Transport to and from the hospital; Care for the patient (by family, friends, and visitors); Isolation, cohorting, or contact precautions. Training of health care professionals and information/communication Legal and insurance costs: Additional insurance costs (patient); Litigation costs, when suing hospitals (patient); Litigation costs, when sued by patients (hospital).	Productivity loss: Loss of productivity/earning/opportunity by the patient due to the resistant infection or sequelae or dying from the resistant infection; Loss of productivity/earnings by family and visitors attending patient; Loss of caretaker (family/friend) productivity. Psychological impact on the patient and family (factored in as QALY). Financial burden on the government for disability benefits. Hospital costs: Reduced patient turnover and decreased revenues (due to longer hospital duration or to isolation/cohorting, or to decision not to perform a non-essential procedure); Reduced capacity of hospital (due to longer hospital duration or to isolation/cohorting) Reputational costs by the hospital. Research and development of new antibiotics. Additional costs non directly related to human health.	Mortality (overall): deaths WITH a MDR infection. Mortality (attributable): deaths FROM an MDR infection. Morbidity: Long term consequences (e.g., chronic or recurrent infections, long-term disability, lower QoL, etc.) from AMR infections; Adverse events, if treated with 2nd-, 3rd-, etc. line drugs. longer hospital (or ICU) stay. Additional diagnostic procedures. Screening programs. Insurance to cover extra AMR costs.

## Data Availability

All the available data are presented in the manuscript.

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
