# Peer review of "State of the Art of Antimicrobial and Diagnostic Stewardship in Pediatric Setting"

_antibiotics, 2025, doi:10.3390/antibiotics14020132_

Round 1

Reviewer 1 Report

Comments and Suggestions for Authors

The abstract should clearly define the scope of the review, such as whether it focuses on the current state of ASPs and DSPs in pediatric care, their implementation challenges, or emerging technologies. Rephrase the objective for precision, e.g., "This review examines the current landscape of antimicrobial and diagnostic stewardship programs (ASPs and DSPs) in pediatric care, highlighting their importance in combating antimicrobial resistance (AMR) and improving health outcomes."

For accessibility, briefly define ASPs and DSPs for readers who may not be familiar with the terms. For example, "ASPs focus on optimizing antimicrobial use to prevent resistance, while DSPs aim to improve diagnostic accuracy to guide appropriate treatment."

The manuscript could benefit from more specific examples or quantitative insights into the cost-effectiveness or impact of ASPs. For instance, mention particular studies, metrics, or outcomes where possible.

Emphasize pediatric-specific challenges or nuances, such as differences in antibiotic prescribing practices between adults and children or the unique role of families in stewardship programs.

The section on future directions and emerging opportunities, such as the use of artificial intelligence, could be more detailed. Suggest specific AI applications (e.g., predictive analytics for resistance patterns) or tools currently under development.

Explicitly state how the findings of this review can influence clinical practice or policy development in pediatric care. For example, "The review highlights practical strategies for implementing ASPs in pediatric settings, emphasizing the need for tailored educational programs and cross-disciplinary collaboration."

The conclusion should be more concise and impactful. Focus on how addressing the identified gaps will directly enhance ASP effectiveness and combat AMR in pediatric populations.

Author Response

The abstract should clearly define the scope of the review, such as whether it focuses on the current state of ASPs and DSPs in pediatric care, their implementation challenges, or emerging technologies. Rephrase the objective for precision, e.g., "This review examines the current landscape of antimicrobial and diagnostic stewardship programs (ASPs and DSPs) in pediatric care, highlighting their importance in combating antimicrobial resistance (AMR) and improving health outcomes."

Thank you for your comment, we changed the abstract

For accessibility, briefly define ASPs and DSPs for readers who may not be familiar with the terms. For example, "ASPs focus on optimizing antimicrobial use to prevent resistance, while DSPs aim to improve diagnostic accuracy to guide appropriate treatment."

Thank you for your suggestion; we added the definition.

The manuscript could benefit from more specific examples or quantitative insights into the cost-effectiveness or impact of ASPs. For instance, mention particular studies, metrics, or outcomes where possible.

Thank you for your suggestion. We added some other information regarding costs to the specific section.

Emphasize pediatric-specific challenges or nuances, such as differences in antibiotic prescribing practices between adults and children or the unique role of families in stewardship programs.

Thank you for pointing this out; we emphasized this part in the introduction section.

The section on future directions and emerging opportunities, such as the use of artificial intelligence, could be more detailed. Suggest specific AI applications (e.g., predictive analytics for resistance patterns) or tools currently under development.

Thank you for your suggestion, we added some information to the discussion section regarding AI.

Explicitly state how the findings of this review can influence clinical practice or policy development in pediatric care. For example, "The review highlights practical strategies for implementing ASPs in pediatric settings, emphasizing the need for tailored educational programs and cross-disciplinary collaboration."

Thank you very much, we added this statement in the conclusion section

The conclusion should be more concise and impactful. Focus on how addressing the identified gaps will directly enhance ASP effectiveness and combat AMR in pediatric populations.

Thank you for pointing this out, we changed the conclusions and we hope that now they are more concise and impactful.

Reviewer 2 Report

Comments and Suggestions for Authors

I congratulate the authors for a narrative review written in a way that is generally consistent with the flow and integrity. However, I have some criticisms. I believe that the manuscript will be improved if this is followed:

It is not clear why the focus is only on microbiomes. I would expect you to discuss the impact of such approaches on other omics.

Line 67: adherence or compliance or both? Please identify which of the terms is more appropriate.

I would expect you to focus on the differences and similarities between ASPs and DSPs definitions in the introduction section.

Line 75: antibiotic or antimicrobial?

Line 120: Nurses not only help with reminding prescriptions, time, duration of treatment and dose, but also play a role in the preparation of drugs, infusion time, stability and incompatibility.

Fig 1: There is also a need for dietitians who examine the interaction of drugs with nutrients and the administration protocol and incompatibility when administered simultaneously and through the same route with NG/OG/PEG/TPN.

Fig 2 has very small fonts that are difficult to read. I recommend adding it as a supplement or converting it to landscape format.

Line 191: In order to prevent information pollution in CDSs such as mobile applications, it is necessary to know who the developer is. For example, it would be beneficial to create them based on the guidelines obtained from national and international professional societies.

Line 195: When educating caregivers, their sociocultural levels should also be taken into account. They should be informed in simple language, especially about vaccines that they think are not appropriate according to their religious beliefs.

Line 212: Which type more appropriate? Electronic or written?

Line 347: Please explain in more detail why a separate heading has been opened for the pediatric emergency department.

Line 640: When evaluating cost-effective analysis, it is useful to emphasize two different approaches: in some countries, the cost is covered entirely by the patient, while in others, it is covered partially/entirely by the government. The importance of social outcomes varies between these two.

Fig 3: In all of the mentioned stages, hospital capacity and workload must be taken into account. For example, how many patients are there per clinician? What are the shift schedules and weekly working hours? Do working hours decrease as the clinical experience of the team increases? At the same time, the patient's treatment complexity is also effective in this process (NICU/PICU vs. other units). For this reason, it is useful to use patient acuity tools (PMID: 37124199).

I would expect you to give more space to POCT genotyping (pMID: 35311942) and AI studies (PMID: 33779743). It is important to support the future approach with concrete studies.

Author Response

I congratulate the authors for a narrative review written in a way that is generally consistent with the flow and integrity. However, I have some criticisms. I believe that the manuscript will be improved if this is followed:

It is not clear why the focus is only on microbiomes. I would expect you to discuss the impact of such approaches on other omics.

Antimicrobial Stewardship Programs (ASPs) specifically impact microbiomes by directly influencing the composition, diversity, and resilience of microbial communities, particularly in pediatric patients. By optimizing antibiotic use, ASPs minimize unnecessary disruption of the microbiome, which is crucial during its developmental stages. This can reduce long-term adverse outcomes such as dysbiosis-related conditions (e.g., allergies, asthma, metabolic disorders). In contrast, other 'omics' approaches, like genomics or proteomics, provide broader systemic insights but are not as directly impacted by antibiotic use as microbiomes. Thus, focusing on microbiomes highlights the immediate, tangible benefits of ASPs in preserving microbial health and preventing cascading effects of dysbiosis. This justifies the targeted analysis within the scope of this study.

Line 67: adherence or compliance or both? Please identify which of the terms is more appropriate.

Thank you for your comment, we changed adherence with compliance

I would expect you to focus on the differences and similarities between ASPs and DSPs definitions in the introduction section.

Thank you for your suggestion; we added a sentence in the part regarding the definition of ASPs and DSPs.

Line 75: antibiotic or antimicrobial?

Thank you for pointing this out, we changed it accordingly.

Line 120: Nurses not only help with reminding prescriptions, time, duration of treatment and dose, but also play a role in the preparation of drugs, infusion time, stability and incompatibility.

Thank you for your comment, we added your suggestion in the specific section.

Fig 1: There is also a need for dietitians who examine the interaction of drugs with nutrients and the administration protocol and incompatibility when administered simultaneously and through the same route with NG/OG/PEG/TPN.

Thank you for your suggestion, we added the figure of dietitians between those that should be included in the t ASP team.

Fig 2 has very small fonts that are difficult to read. I recommend adding it as a supplement or converting it to landscape format.

Thank you for bringing this to our attention. We have replaced Figure 2 with a table to improve readability.

Line 191: In order to prevent information pollution in CDSs such as mobile applications, it is necessary to know who the developer is. For example, it would be beneficial to create them based on the guidelines obtained from national and international professional societies.

Thank you for pointing this out, we clarified this aspect.

Line 195: When educating caregivers, their sociocultural levels should also be considered. They should be informed in simple language, especially about vaccines that they think are inappropriate according to their religious beliefs.

Thank you for your comment; we added it to the caregiver’s education part.

Line 212: Which type more appropriate? Electronic or written?

Thank you for pointing this out, we clarified this aspect in the specific paragraph.

Line 347: Please explain in more detail why a separate heading has been opened for the pediatric emergency department.

Thank you for your comment, we made more explicit the reasons why PEDs is considered a hybrid setting.

Line 640: When evaluating cost-effective analysis, it is useful to emphasize two different approaches: in some countries, the cost is covered entirely by the patient, while in others, it is covered partially/entirely by the government. The importance of social outcomes varies between these two.

Thank you for pointing this out, we agreed with you, and we added a specific section in the cost-effectiveness part.

Fig 3: In all of the mentioned stages, hospital capacity and workload must be taken into account. For example, how many patients are there per clinician? What are the shift schedules and weekly working hours? Do working hours decrease as the clinical experience of the team increases? At the same time, the patient's treatment complexity is also effective in this process (NICU/PICU vs. other units). For this reason, it is useful to use patient acuity tools (PMID: 37124199).

Thank you very much, we agreed with you. We specified better this part in section 10.

I would expect you to give more space to POCT genotyping (pMID: 35311942) and AI studies (PMID: 33779743). It is important to support the future approach with concrete studies.

Thank you for your suggestion, we added more information about these topics in the discussion section.

Round 2

Reviewer 2 Report

Comments and Suggestions for Authors

Thank you for the revised version of manuscript. It is suitable for publication.